# Video Face Re-Aging: Toward Temporally Consistent Face Re-Aging

## Abstract

Video face re-aging deals with altering the apparent age of a person to the target age in videos. This problem is challenging due to the lack of paired video datasets maintaining temporal consistency in identity and age. Most re-aging methods process each image individually without considering the temporal consistency of videos. While some existing works address the issue of temporal coherence through video facial attribute manipulation in latent space, they often fail to deliver satisfactory performance in age transformation. To tackle the issues, we propose (1) a novel synthetic video dataset that features subjects across a diverse range of age groups; (2) a baseline architecture designed to validate the effectiveness of our proposed dataset, and (3) the development of novel metrics tailored explicitly for evaluating the temporal consistency of video re-aging techniques. Our comprehensive experiments on public datasets, including VFHQ and CelebV-HQ, show that our method outperforms existing approaches in age transformation accuracy and temporal consistency. Notably, in user studies, our method was preferred for temporal consistency by 48.1% of participants for the older direction and by 39.3% for the younger direction.

## 1 Introduction

Video face re-aging or video-based face re-aging aims to transform the apparent age in facial videos while ensuring the temporal consistency in both age and identity. This field holds significance relevance across diverse domains, including computer graphics, forensics, entertainment, and advertising. Despite the extensive research conducted in this domain, the challenge remains largely unexplored when it comes to videos. One of the remaining challenges is that existing image-based methods yield inconsistent identities when applied to videos or consecutive frames featuring varying expressions, viewpoints, and lighting conditions.

Seminal studies Alaluf et al. (2022); Tzaban et al. (2022) have leveraged StyleGAN-based Karras et al. (2019) frameworks to develop techniques for manipulating facial attributes in videos, aiming for greater attribute consistency. Zoss et al. (2022) have utilized the synthetic images labeled with existing re-aging techniques Alaluf et al. (2021) to curate a paired dataset for re-aging. Their study shows that supervised training on synthetic dataset yields favorable outcomes for still images. Preechakul et al. (2022); Chen & Lathuilière (2023); Li et al. (2023); Wahid et al. (2024) introduce diffusion models to tackle this problem. Lin et al. (2024) employs a reward-based approach by establishing a consensus between the aging process and the aging personalization agents to generate robust faces. However, these approaches are trained on static images and significantly suffer from temporal inconsistencies.

Thus, training face re-aging methods on videos is beneficial for addressing the temporal consistency problem in video re-aging tasks. Therefore, we propose a pipeline to generate a synthetic video dataset. This dataset comprises paired data for supervised training, consisting of various ages, poses, and expressions. The creation of this dataset involves three major steps.

Firstly, we utilize StyleGAN to synthesize a face image. Then we apply existing re-aging method SAM Alaluf et al. (2021) to generate images of the same person with varying ages (Sec. 3.1.1). Next, we build key frames consisting of various poses and expressions of that individual (Sec. 3.1.2). Lastly, we introduce natural motion with frame interpolation (Sec. 3.1.3).

Figure 1: Our proposed pipeline to construct the video dataset for re-aging. Firstly, high-resolution synthetic facial images are created using StyleGAN Karras et al. (2019) Subsequently, images of individuals at different target ages are generated using SAM Alaluf et al. (2021) for age transformation. Next, key frames are produced by employing OSFV, which alters the pose and expression of these synthetic images. This is achieved without relying on driving images, instead using random values for rotation, translation, and expression keypoints. Finally, motion is added to these key frames using FILM Reda et al. (2022), creating smooth and high-fidelity motion videos of subjects at different ages.

In addition to the dataset, we introduce a baseline architecture designed to utilize the temporal coherence inherent in our proposed video dataset. This architecture primarily comprises recurrent blocks, employing a fusion-based approach that leverages concatenated inputs to exploit temporal consistency. Drawing inspiration from seminal works in video generation Clark et al. (2019); Saito et al. (2020); Tulyakov et al. (2018), we incorporate a video discriminator equipped with 3D convolutional layers to ensure both realism and natural motion in the generated videos.

Recognizing that existing aging metrics are not well-suited for video-based methods, we address this gap by developing novel metrics to assess temporal continuity in video-based re-aging. Through extensive experiments, we demonstrate that our video-based architecture produces remarkable results and outperforms existing state-of-the-art methods across various public datasets.

Given the challenges and limitations of current methods, our work introduces several contributions in video re-aging as follows.

1. We introduce a pipeline designed to generate a synthetic video dataset specifically for video re-aging. This dataset features age-paired videos of individuals across various ages, poses, and expressions, allowing model to be trained with supervised learning.

2. We present a baseline network architecture custom-designed for our synthesized video dataset. Our generator is built upon a combination of recurrent blocks consisting of U-Net architecture, utilizing both 2D image-based and 3D video-based conditional discriminators.

3. We propose novel metrics to evaluate the temporal consistency of video re-aging methods. These include Temporal Regional Wrinkle Consistency (TRWC) and Temporally Age Preservation. These metrics provide a robust framework for evaluating the quality of age transformations over time.

## 2 RELATED WORKS

### 2.1 IMAGE-BASED FACE RE-AGING

The study in Antipov et al. (2017b) pioneered the use of a conditional GAN for face aging. Subsequently, several influential works such as Antipov et al. (2017a); Wang et al. (2018); Or-El et al. (2020); Li et al. (2021); Yao et al. (2021b); Kim et al. (2024); Guo et al. (2024) emerged, expanding on this concept. Similar to Li et al. (2021), SAM Alaluf et al. (2021) emphasizes continuous age progression. SAM is a StyleGAN-based model Karras et al. (2019) capable of generating high-resolution images. In contrast to other methods, it does not use an age classifier to estimate the input age. FRAN Zoss et al. (2022) trains a simple encoder-decoder network in a supervised manner with a generated synthetic dataset. Rather than adopting age classifiers or embedding, it extends the input to a 5-channel image, including two binary masks for input and target ages. CUSP Gomez-Trenado

et al. (2022) introduces style and content encoders to disentangle the style and content of the input. This approach also incorporates the GB algorithm Springenberg et al. (2014) within the CUSP module, ensuring that only age-relevant features are processed. AgeTransGAN Hsu et al. (2022) disentangles the encoded image into identity and age components with two independent modules. PADA Li et al. (2023) and FADING Chen & Lathuilière (2023) adopt a text-driven approach and integrates the pre-trained CLIP Radford et al. (2021) and diffusion models. However, these works only focus on images, and directly applying these methods on individual frames does not consider temporal consistency, affecting the quality of re-aging quality over time.

## 2.2 VIDEO-BASED FACE RE-AGING

Most video face re-aging methods transform the face by manipulating the age in latent space, except Duong et al. (2019) that proposes a reinforcement learning method for the sequence of video frames. Yao et al. (2021a) proposes editing the vectors in the StyleGAN latent space with various pre-processing steps that are independently applied to every single frame of an input video. Tzaban et al. (2022) identify the inconsistencies in PTI Roich et al. (2022) and suggest e4e Tov et al. (2021) encoder with it for finding the pivots. Video editing methods often crop faces as a prepossessing step. To overcome this problem, Yang et al. (2023) addresses this issue by proposing changes in the initial StyleGAN layers to overcome the cropping problem. Kim et al. (2023) proposes a diffusion-based editing approach by using Preechakul et al. (2022) that disentangles the video into time-dependent features (such as motion) that are applied to each frame and time-independent features (such as identity) which are shared across all the frames. While improving temporal consistency over image-based methods, they still struggle to accurately transform faces to the target age.

t

## 2.3 VIDEO FACE RE-AGING DATASET

Video face re-aging presents significant challenges, primarily due to the lack of dedicated video datasets. Existing image-based face re-aging datasets are typically labeled either automatically using an age classifier or manually via crowdsourcing Zhang et al. (2022); Liu et al. (2021); Karras et al. (2019); Rothe et al. (2018). However, these datasets do not provide paired ages for supervised training. Zheng et al. (2017) proposed a technique to generate a synthetic dataset that is labelled through SAM Alaluf et al. (2021). Duong et al. (2019) also labelled a video dataset, which remains private.

## 3 VIDEO FACE RE-AGING FRAMEWORK

We first describe the pipeline of synthesizing the proposed video dataset. This is followed by introducing our baseline architecture and loss functions. Lastly, we present novel metrics specifically designed for video re-aging methods to quantify their re-aging performance over time.

### 3.1 RE-AGING VIDEO DATASET

#### 3.1.1 IMAGE-BASED FACE RE-AGING DATASET

Creating a high-quality image-based re-aging dataset is a crucial in our pipeline because it directly influences the quality of our subsequent video re-aging dataset. However, obtaining these paired image datasets is a challenging task. To overcome this limitation, we turn to insights from Zoss et al. (2022), which show that training on synthetic datasets can yield realistic results on real images. For instance, the neural network can learn how wrinkles change from the synthetic images and apply this knowledge to real images. We refer to these learned changes as delta images $D_t$, as illustrated in Fig. 2.

Firstly, we leverage StyleGAN Karras et al. (2020) which takes a random noise as input and generate high-resolution synthetic image. Then we utilize SAM Alaluf et al. (2021) that manipulates the latent vector with StyleGAN for age transformation. The intuition behind choosing SAM over other existing methods is its re-aging performance in terms of age error, that is also evident through our experiments. Using this approach, we construct a synthetic facial image dataset:

$$I^{tar} = SAM(G_z(z); a^{tar}) \tag{1}$$

Given a random noise $z$, StyleGAN $G_z(\cdot)$ produces a sample image, which is then used as input for SAM Alaluf et al. (2021) along with a target age $a^{tar}$. As a result we get an image $I^{tar}$ with apparent age $a^{tar}$ as shown in Fig. 1 (a). Readers can refer to Zoss et al. (2022) for more details about image-level re-aging process.

### 3.1.2 KEY FRAMES GENERATION

The subsequent step in the outlined pipeline involves key frames generation. This key frames capture specific snapshots or moments in videos to enable seamless transitions between sequences. The challenge lies in acquiring diverse facial images to adequately encapsulate the video's dynamics, including various poses and expressions. To address this, we utilize a recent face reenactment technique Wang et al. (2021a) that modifies the pose and expression of a source image based on a driving image. Incorporating this method into our pipeline allows us to generate multiple images of an individual with varying poses and expressions. These resulting images serve as key frames.

In this study, we employ the off-the-shelf model OSFV[1]Wang et al. (2021a) to produce synthetic key frames using our image dataset. Our approach diverges from the original methodology in that we rely solely on source images from our dataset. Instead of utilizing driving images, which can be challenging to gather, we employ random values for the rotation matrix $R$, translation matrix $Q$, and expression keypoints $\delta$ to generate various poses and expressions as shown in Fig. 1 (b). We can write this process as follows:

$$K = G_{kp}(I^{tar}, R, Q, \delta), \tag{2}$$

where $K$ represents generated key frames and $G_{kp}$ denotes key frames generator. We generate eight different key frames through this method. While it is also possible to repeat this procedure to create a motion video, but we have observed a degradation of the quality in the resultant videos. Therefore, we come up with an alternative approach to address this problem in next section.

### 3.1.3 MOTION GENERATION

The final step of our pipeline is motion generation. We leverage the recent work in frame interpolation methods to ensure smooth and high-fidelity motion. Specifically, we employ the method presented in Reda et al. (2022) to the eight key frames generated in the previous step by recursively generating intermediate frames between them. This iterative process is executed for two consecutive frames and is repeated for every subject across all ages as follows:

$$I_t = FI(I_{t-1}, I_{t+1}), \tag{3}$$

where $FI$ is motion generation network Reda et al. (2022) and $I_t$ is the frame at $t$ time-step. As a result, we obtain smooth and high-fidelity paired videos for every subject of different ages as shown in Fig. 1 (c).

## 3.2 NETWORK ARCHITECTURE

### 3.2.1 GENERATOR

To fully utilize the advantage of our novel video re-aging dataset, we carefully design the generation scheme to accurately transform faces over time. Following the approach in Zoss et al. (2022), we adopt an alternative method for incorporating input and target ages into our model structure. Recent methods Gomez-Trenado et al. (2022); Hsu et al. (2022) tend to use pre-trained age classifiers to estimate the input age. We concatenate the input frame $I_t$ at $t$ time-step with two spatial masks (one for input age and one for target age) over channel dimensions. These two masks contain constant

---

[1]Note that we trained OSFV on VFHQ Xie et al. (2022) dataset for a resolution of $512\times512$. The training takes 30 days on 8 A100 GPUs. Additional details are provided in the supplementary materials.

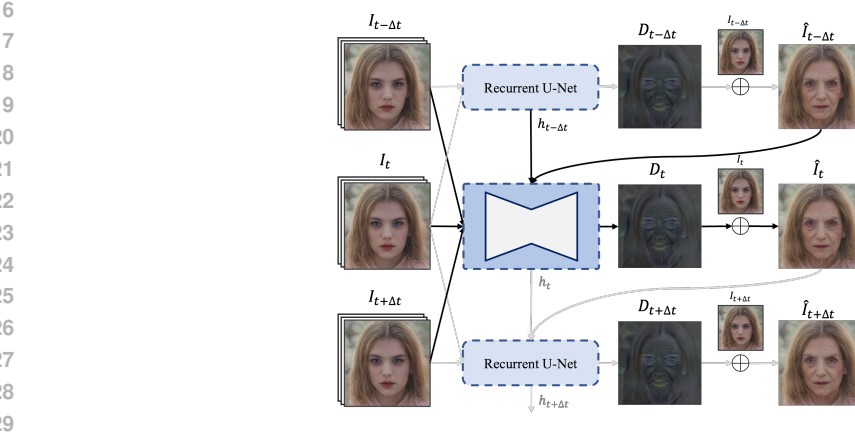

Figure 2: Overview of our generator for video re-aging.

values representing input and output ages, ranging from 0 to 100 normalized between 0 and 1. This results in a 5-channel masked frame $I_t^{mask}$. Mathematically,

$$I_t^{mask} = [I_t, M^{inp}, M^{tar}], \tag{4}$$

where $M^{inp}$ and $M^{tar}$ are spatial masks for input and target ages. $[\cdot]$ denotes the channel-wise concatenation operation. If $V^{mask} = \{I_1^{mask}, I_2^{mask}, \ldots, I_N^{mask}\}$ refers to any input masked sequence $V^{mask}$ for $N$ total number of frames, our generator $G$ produces the output video $\hat{V}$.

$$\tilde{V} = G(V^{mask}), \tag{5}$$

The proposed $G$ adopts a recursive scheme to consider the temporal information of videos, inspired by Wu et al. (2022); Tian et al. (2021); Zhu et al. (2023), namely recurrent block function $RB(\cdot)$. We employ the U-Net architecture within the $RB$ function. Please refer to Sec. A for the details of U-Net network architecture. This $RB$ stacks the multiple arguments. First, we concatenate the consecutive input frames $[I_{t-\Delta t}^{mask}, I_t^{mask}, I_{t+\Delta t}^{mask}]$. Here, $I_{t-\Delta t}^{mask}$ and $I_{t+\Delta t}^{mask}$ refers to the two adjacent frames of $I_t^{mask}$ with interval step $\Delta t$. We further concatenate the resulting stacked frames with previous hidden state $h_{t-\Delta t}$ and previous output frame $\hat{I}_{t-\Delta t}$. Mathematically,

$$h_t, D_t = RB([I_{t-\Delta t}^{mask}, I_t^{mask}, I_{t+\Delta t}^{mask}, \tilde{I}_{t-\Delta t}, h_{t-\Delta t}]) \tag{6}$$

As a result, we obtain hidden state $h_t$ and delta image $D_t$. Once the delta image is attained, we can easily obtain output frame $\hat{I}_t$ through element-wise summation between input $I_t$ and $D_t$:

$$\hat{I}_t = D_t + I_t. \tag{7}$$

We have now processed three frames to get the re-aged output of the middle frame. For the remaining consecutive frames, we can pass the hidden state $h_t$ and output frame $\hat{I}_t$ along with the next consecutive frames in Eq. 6 for the next iteration. This process is repeated for all $N$ frames of the input video $V^{mask}$ to generate the output video $\hat{V}$, as illustrated in Fig. 2.

### 3.2.2 DISCRIMINATOR

In addition to the image discriminator, we introduce a video discriminator to assess the consistency of age-related high frequency details across consecutive frames. We observed that using only image discriminator produces inconsistent output, leading to flickering. The comparison is presented in our ablation studies. The image discriminator employs a PatchGAN architecture Isola et al. (2017) to differentiate realistic images from synthetic ones, while for the video discriminator, we adopt a

Figure 3: Conceptual overview of proposed TRWC.

spatio-temporal convolutional network that utilizes 3D convolution layers, similar to the approach used in previous works such as Vondrick et al. (2016) and Tulyakov et al. (2018). Specifically, we exploit the spatio-temporal information from consecutive generated frames in the form of delta images $D_t$ as inputs to important details details over time without depending on facial images.

### 3.2.3 LOSS FUNCTIONS

We use $L1$ loss, LPIPS loss Zhang et al. (2018), and adversarial loss functions Lim & Ye (2017). We also use adversarial loss functions for image and video discriminators $\mathcal{L}_{adv,I}$ and $\mathcal{L}_{adv,V}$. Our total objective is obtained as:

$$\mathcal{L} = \lambda_{L1}\mathcal{L}_{L1}(\hat{V}, V_{gt}) + \lambda_{adv,V}\mathcal{L}_{adv,V}(\hat{V}, M^{tar}) + \lambda_{adv,I}\mathcal{L}_{adv,I}(\hat{V}, M^{tar}) + \lambda_p\mathcal{L}_{LPIPS}(\hat{V}, V_{gt}),$$

where $V_{gt}$ is the ground-truth video corresponding to the target age obtained from dataset building pipeline. Here, we set the lambda values as $\lambda_{L1} = 1.0$, $\lambda_{adv,I} = 0.025$, $\lambda_{adv,V} = 0.025$, $\lambda_p = 1.0$.

### 3.3 PROPOSED METRICS

Evaluating age transformation performance in a video context is both essential and challenging. According to the best of our knowledge, there is no specific metric to assess the temporal consistency of re-aging methods. Most existing evaluation metrics are designed for image-based re-aging methods as they measure the age transformation performance on individual images Zoss et al. (2022) or use metrics designed to evaluate the continuity of identity drift Alaluf et al. (2022). Regarding this, we propose two metrics, TRWC and T-Age, capable of effectively assessing temporal continuity in terms of age in video-based re-aging task.

### 3.3.1 TEMPORAL REGIONAL WRINKLE CONSISTENCY (TRWC)

Te evaluate the consistency of video re-aging methods, we need to consider the aging-related details in facial areas such as wrinkles where it matters. This is because these regions are most vulnerable and simultaneously important in terms of human perception. Therefore, we focus on aging-related facial areas, especially Crow's feet and Nasolabial folds, motivated from Wang et al. (2021b); Li et al. (2022). These areas are more likely to show signs of aging because they are used a lot for facial expressions and speech. We extract the region-of-interest (ROI) around these areas to observe changes over time. However, when analyzing perceptual differences between frames, factors such as facial angles and expressions can be sensitive. To address this, we calculate the LPIPS of the synthesized image and normalize it using the LPIPS of the real image, ensuring we consider differences in each image over time. Fig. 3 explains the conceptual schematic of TRWC. Mathematically, we define TRWC as follows.

$$\text{TRWC}_{\Delta t} = \frac{1}{3(N - \Delta t)} \sum_{r=1}^{3} \sum_{t=1}^{N-\Delta t} \frac{lpips(f_{roi}^r(\hat{I}_t), f_{roi}^r(\hat{I}_{t+\Delta t}))}{lpips(f_{roi}^r(I_t), f_{roi}^r(I_{t+\Delta t}))}, \tag{8}$$

Table 1: Quantitative comparison on CelebV & VFHQ datasets. The best results are highlighted in bold.

| Dataset | Models | Young $\to$ Old | | Old $\to$ Young | |
|---|---|---|---|---|---|
| | | TRWC$_1\downarrow$ | T-Age$\downarrow$ | TRWC$_1\downarrow$ | T-Age$\downarrow$ |
| CelebV-HQ | AgeTransGAN | 4.25 | 1.26 | 3.12 | 2.29 |
| | CUSP | - | - | 1.90 | 2.19 |
| | FADING | 4.21 | 3.20 | 2.23 | 3.14 |
| | **OURS** | **1.38** | **0.84** | **0.70** | **1.03** |
| VFHQ | AgeTransGAN | 4.26 | 1.61 | 2.92 | 2.04 |
| | CUSP | - | - | 1.76 | 2.14 |
| | FADING | 3.79 | 2.77 | 2.04 | 2.47 |
| | **OURS** | **1.35** | **1.16** | **0.69** | **1.34** |

where $N$ is the number of frames and $\Delta t$ is time interval between consecutive frames. The function $f_{roi}^r(\cdot)$ is ROI function that are obtained by facial landmark detector Deng et al. (2018). Here, we consider only three ROI regions, left-eye, right-eye, and mouth, indexed by $r$.

### 3.3.2 TEMPORAL-AGE (T-AGE) PRESERVATION

Drawing inspiration from the TL-ID metric Tzaban et al. (2022), which proposes metrics for identity consistency in videos, we introduce T-Age for video re-aging. T-Age measures the age difference between two adjacent frames using cosine similarity, utilizing an off-the-shelf age classifier Rothe et al. (2015). A lower T-Age value indicates a more consistent age representation across the frames.

## 4 EXPERIMENTS

### 4.1 EXPERIMENT SETUP

In this section, we show the superiority of our method by comparing with all existing state-of-the-art re-aging methods. This includes HRFAE Yao et al. (2021b), SAM Alaluf et al. (2021), FRAN Zoss et al. (2022), CUSP Gomez-Trenado et al. (2022), AgeTransGAN Hsu et al. (2022), Diffusion AE Preechakul et al. (2022), STIT Tzaban et al. (2022), StyleGANEX Yang et al. (2023), and Diffusion VAE Kim et al. (2023), FADING Chen & Lathuilière (2023) totaling 10 methodologies.

We have trained our methods on the proposed synthetic videos generated through our pipeline. We have shared the sample images of our generated dataset in Fig. 7. For the test set, we choose the CelebV-HQ Zhu et al. (2022) and VFHQ dataset Xie et al. (2022) as video test set. We consider three target age groups, $(18, 25, 35)$ with input age as 85 for $Old \to Young$ task and three age groups, $(65, 75, 85)$ with input age as 18 for $Young \to Old$ task. We provide the additional details in the supplementary materials (Sec. A).

### 4.2 METRICS

We evaluate the performance of re-aging models based on their ability to transform to the target age while ensuring temporal consistency in the re-aged image. Given that we do not have access to ground-truth for real videos, we employ metrics that do not rely on ground-truth. Specifically, we use five metrics to evaluate the results. We calculate mean absolute error (MAE) between the estimate ages, computed with the pre-trained age classifier to quantify the age transformation quality whereas temporal consistency is measured by TRWC and T-Age. Despite knowing that many existing methods employ DEX that might bias the test results, we opted for DEX due to its accuracy and stability. However, it's important to note that we did not use any age estimation network in our training process. Lastly, we use TGID Tzaban et al. (2022) to evaluate the global identity similarity.

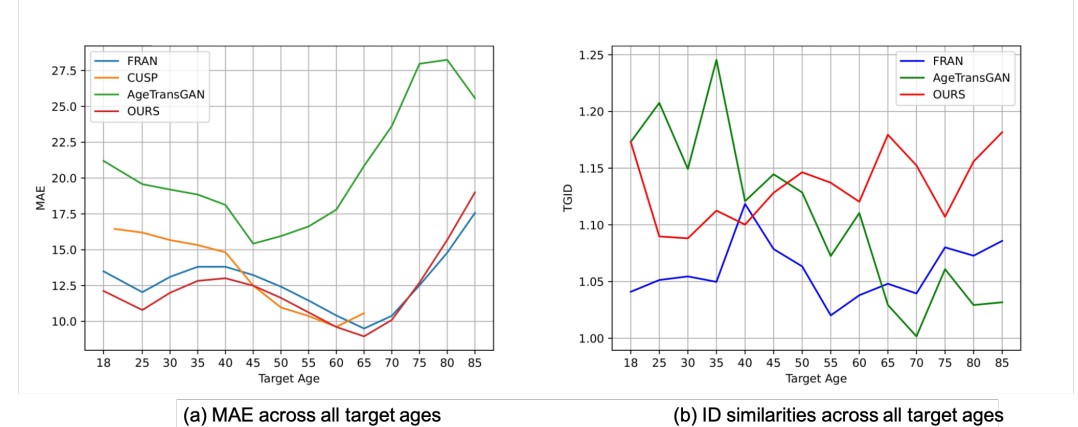

(a) MAE across all target ages        (b) ID similarities across all target ages

Figure 4: Comparison of (a) Mean Absolute Error (MAE) and (b) ID Similarity (TGID) across a range of target ages for different methods. Lower MAE and higher TGID values indicate better performance.

Table 2: User Study Results. Participants evaluated the methods according to four criteria: Age Accuracy (**AA**), Identity Preservation (**IP**), Temporal Consistency (**TC**), and Overall Naturalness (**ON**). The highest-scoring results are highlighted in bold. Scores are in percentages (**%**).

| Models | Young → Old | | | | Old → Young | | | |
|---|---|---|---|---|---|---|---|---|
| | AA↑ | IP↑ | TC↑ | ON↑ | AA↑ | IP↑ | TC↑ | ON↑ |
| AgeTransGAN | 10.56 | 11.80 | 11.14 | 12.38 | 22.39 | 31.83 | 27.06 | 25.23 |
| CUSP | 21.81 | **46.07** | 21.02 | 29.02 | 12.56 | 14.31 | 11.82 | 12.85 |
| **OURS** | **67.64** | 42.13 | **67.84** | **58.60** | **65.05** | **53.86** | **61.12** | **61.92** |

## 5 COMPARISON RESULTS

### 5.1 QUANTITATIVE RESULTS

We quantitatively compare our method with state-of-the-art methods in Table 1. Note that we omit the CUSP result for $Young \rightarrow Old$ due to its maximum target age limitation (65) Gomez-Trenado et al. (2022). Our results indicate that model trained on videos show higher TRWC, suggesting that it maintains greater temporal stability and exhibits lower perceptual differences between the consistent frames. On other side, When evaluating age transformation continuity using T-AGE, our method outperforms the other methods. Furthermore, it shows that diffusion models, i.e., FADING Chen & Lathuilière (2023), are also temporally inconsistent. These results suggest that these metrics' performances are strongly correlated with the target ages. Therefore, we show the MAE for all the target ages in Fig. 4 (a). Furthermore, conserving the identity is important when changing the age. Therefore, by following Tzaban et al. (2022), we compare the identity similarity in Fig. 4 (b). The trends show that our methods preserve the identity when changing the ages, even for older ages, as other datasets are skewed towards younger ages. We can observe that the age transformation performance of our method is consistently better than other state-of-the-art methods, despite being trained on synthetic videos. Additionally, our method also improves overall temporal consistency. It is worth mentioning that all the results we presented are in line with user studies (Table. 2), which further affirm our newly proposed metrics.

### 5.2 USER STUDY

We conducted a subjective evaluation via a user study, presenting 15 sequences to a total of 43 anonymous participants for the older direction and 36 participants for the younger direction. This study compares our method with state-of-the-art techniques such as CUSP Gomez-Trenado et al. (2022), and AgeTransGAN Hsu et al. (2022) on $Young \rightarrow Old$ and $Old \rightarrow Young$ tasks. As illus-

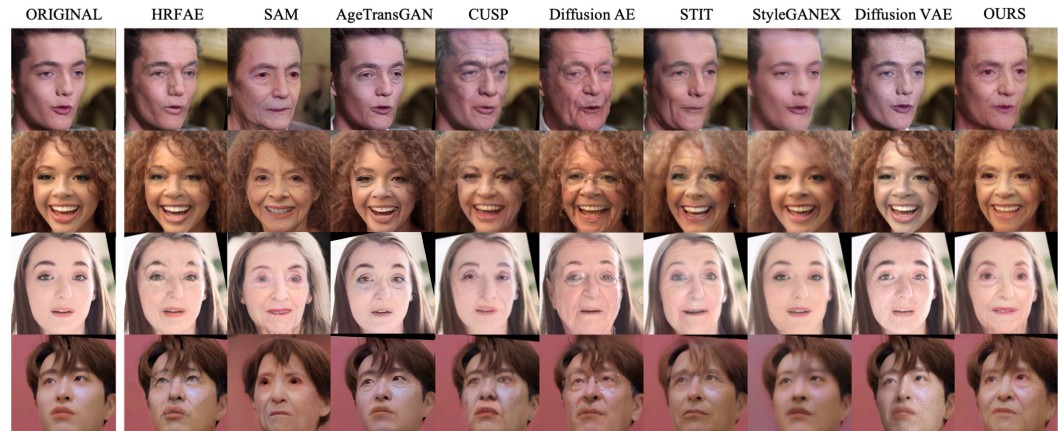

Figure 5: Qualitative comparison with existing state-of-the-art methods. The target age is set to 85.

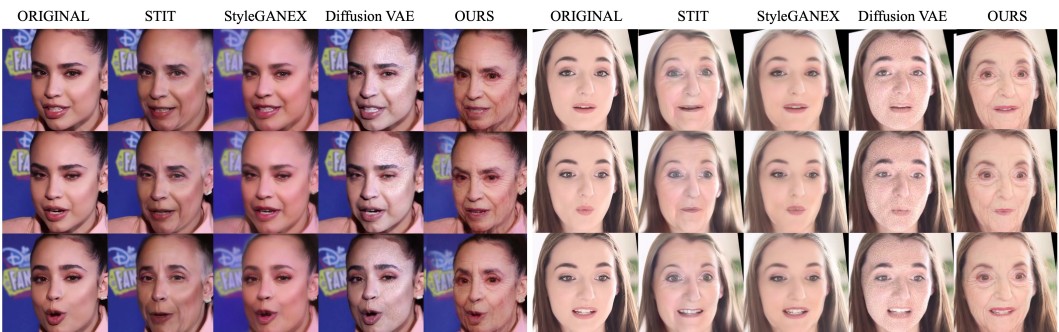

Figure 6: Comparison with video editing methods.

trated in Table. 2, a clear majority of participants found our method superior, particularly in terms of age accuracy, temporal consistency, and naturalness. These findings highlight the effectiveness of our video-based approach in enhancing both temporal consistency and age transformation capabilities, thereby demonstrating that proposed metrics are reliable indicators of cognitive temporal consistency. For example, the significance of a 0.18 (±1) difference in TRWC in Table. 1 is evident by the users' choices.

## 5.3 QUALITATIVE RESULTS

We present our qualitative comparison with the existing state-of-the-art in Fig. 5. It is evident that SAM is effective in age transformation, while STIT and StyleGANEX struggle in this aspect. However, SAM fails to preserve attributes such as identity, pose, and expression because it strongly depends on age classifier loss, which is proficient at altering age but not at maintaining other key attributes. On the other hand, STIT and StyleGANEX, which utilize average vectors related to age changes, fail to properly consider the individual samples. As a result, they encounter difficulties in age transformation, particularly when dealing with wild samples. In such cases, their methods become more susceptible due to their strong reliance on trained data for constructing the latent space. Diffusion AE achieves successful age transformation but introduces artifacts such as glasses, leading to identity drift. While the Diffusion VAE works for small changes, it fails entirely in age transformation when dealing with a significant age gap for all subjects. Image-based methodologies, such as HRFAE and AgeTransGAN, inherently exhibit lower age transformation capabilities in wild scenarios. In contrast, while CUSP achieves successful age transformation in certain samples, it often tends to produce artifacts. In contrast, our method successfully performs age transformation while maintaining stable results in terms of identity, expression, and pose. While FRAN also achieves sim-

Table 3: Ablation studies: We ablate the (Left) key frame generation and (Right) different frame interpolation methods to select the highest performance method based on CPBD and warping error.

| Method | MAE↓ | | ID↑ | |
|--------|------|------|------|------|
| | 18 | 85 | 18 | 85 |
| StyleHEAT | 8.06 | 4.41 | 0.64 | 0.50 |
| StyleMask | 7.23 | 9.98 | 0.61 | 0.37 |
| OSFV | **1.62** | **0.71** | **0.90** | **0.90** |

| Methods | DQBC | EMA-VFI | FILM |
|---------|------|---------|------|
| CPBD↑ | 0.5115 | 0.4213 | **0.6061** |
| Warp Error↓ | 0.00118 | 0.00119 | **0.00104** |

ilar quality, our results possess more natural wrinkle details, with the differences being particularly pronounced around the eyes.

Furthermore, Fig. 6 presents a performance comparison of different frames to demonstrate the efficacy of our approach against other video editing methods. It can be observed that, despite producing temporally consistent results, their ability to transform age is quite limited. Overall, these comparison results indicate that leveraging a video dataset enables our method to outperform the existing methods, yielding natural age progression and high-quality results, even under extreme conditions. We share more results in Sec. B and discuss the limitations of our work in D.

## 6 ABLATION STUDY

In this section, we ablate the selection of the modules of proposed pipeline except the choice of SAM Alaluf et al. (2021) which is already validated in Zoss et al. (2022).

### 6.0.1 KEY FRAME GENERATION

In Table 3 (a), we compare the performance of three reenactment methods: OSFV Wang et al. (2021a), StyleMask Bounareli et al. (2023), and StyleHeat Yin et al. (2022), for key frame generation. The best method is selected for our re-aging task based on lower MAE and higher identity similarity (ID). For ID, we calculate the cosine similarity between the vectors of the face image, extracted using ArcFace Deng et al. (2019). Given different pose and expressions, it is crucial to maintain the age of a person without losing its identity. The results show that OSFV Wang et al. (2021a) successfully preserves both the identity and age of the person, while other methods fail to maintain the person's identity and age. We further investigate the training configurations of Wang et al. (2021a) in Sec. C.

### 6.0.2 MOTION GENERATION

In this ablation study, we experiment with state-of-the-art frame interpolation models Reda et al. (2022); Zhang et al. (2023b); Zhou et al. (2023) to determine the most effective motion generation method. We evaluate image quality and motion consistency using non-reference-based metrics such as cumulative probability of blur detection (CPBD) Narvekar & Karam (2009; 2011) for image quality and warping error Zhang et al. (2023a) for motion consistency. Table 3 (b) suggests that all methods exhibits lower error in terms of temporal consistency. However, FILM Reda et al. (2022) outperforms other models by producing sharp results, while the outputs of the remaining models tend to be blurry.

## 7 CONCLUSION

In this paper, we introduced a paired video dataset for the video re-aging. This dataset encompasses subjects from a wide range of age groups. Shifting from the conventional model-centric focus, we adopted a data-centric approach. Inspired by recent advancements in face reenactment and frame interpolation, we encompasses various facial poses and expressions in the proposed data. Consequently, we proposed a baseline network architecture to evaluate the proposed dataset, emphasizing both temporal consistency and the quality of age transformations. Furthermore, we formulated two novel metrics to evaluate temporal consistency in video re-aging, which consider age-relevant features such as facial wrinkles over time. We validated our method with comprehensive experiments.

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
