# OpenReview forum: "Video Face Re-Aging: Toward Temporally Consistent Face Re-Aging"
_ICLR.cc/2025/Conference — Submitted to ICLR 2025_

### Official Review · Reviewer_azHF · 2024-11-03

**Soundness:** 2
**Presentation:** 3
**Contribution:** 2
**Rating:** 3
**Confidence:** 5

**Summary:**

In this paper, authors focus on video face re-aging task considering the temporal consistency. Most re-aging methods processed each image individually without integrating temporal dimension of videos due to the lack of paired video datasets for supervised training. Thus, an important contribution from authors is a novel synthesis video dataset created via proposed pipeline, it features many subjects with covering a diverse range of age groups.  Then, a baseline video face re-aging architecture is designed to validate the effectiveness of the proposed video dataset. Last but not least, two tailored novel metrics are developed for evaluating the temporal consistency of video face re-aging task.

**Strengths:**

10 existing state-of-the-art re-aging methods are compared in order to validate the efficacy of proposed synthesis video dataset and baseline architecture on public datasets, such as VFHQ and CelebV-HQ, as well as necessary ablation experiments. The paper is overall well written.

**Weaknesses:**

Although, a new paired video face re-aging dataset is essential for enhancing face re-aging technique and motivating relevant community. Overall, lack of novelty is disadvantage of this manuscript. First, video face re-aging dataset is constructed by a pipeline with three stages. Each of them focuses on off-the-shelf method, such as Style-based Age Manipulation (SAM) is chosen for image-based face re-aging, OSFV is chosen for key frame generation and FILM is chosen for motion generation. It is a general pipeline for constructing video dataset. Second, the proposed baseline architecture of video face re-aging is composed of off-the-shelf building block stacks. Such as recurrent block (RB) and Unet-based Encoder-Decoder. Even the input fashion of the proposed architecture is borrowed from Zoss et al, such as 5 channels with age masks, let alone the discriminator with PatchGAN proposed by Isola et al. Last but not least, the proposed Temporal-Age (T-Age) metric measures the age difference between two adjacent frames utilizing an off-the-shelf age classifier from Rothe at al.  In a short, this manuscript can be considered as a regular technical report, it has a gap to meet the novelty requirement for acceptance.

**Questions:**

There are some questions need to be clarified from authors.
1. In line 292, for image and video discriminator loss , how to explain there is no ground truth in total objective function when updating the discriminator loss ?
2. In Table 1,  three image-based face re-aging methods are compared, is there no comparison with SAM? and how about video-based method, such as diffusion autoencoders (Preechakul et al.) ?
3. In Figure 4 (b), how to explain there is no CUSP results ?
4. In Table 2, how to explain there is no video-based face re-aging method in user study ?
5. In line 468, please give more detailed explanation about the sentence “ the significance of a 0.18 in TRWC in Table. 1 is evident by the user’s choices”
6. Overall, I can’t find more detailed meta information about the proposed video face re-aging dataset, such as how many identities or subjects, total duration of dataset and so on.

**Details Of Ethics Concerns:**

Although, StyleGAN is utilized to generate fake(not exist in real world) face image from a random noise, the proposed dataset may have the potential biases inherited from StyleGAN trained on FFHQ dataset.

---

### Official Review · Reviewer_Lhji · 2024-11-03

**Soundness:** 3
**Presentation:** 3
**Contribution:** 3
**Rating:** 5
**Confidence:** 5

**Summary:**

The paper presents a simple GAN-based approach to generate a video of a subject at the target age. To maintain temporal consistency, the generator employs a recurrent architecture with U-Net blocks. This structure leverages both previous hidden states and generated frames, ensuring smooth transitions between ages. The model is trained using a combination of image and video discriminators, enhancing realism and temporal coherence.  Furthermore, the authors develop a pipeline for generating synthetic aging datasets and propose two new metrics for evaluating the temporal consistency of video re-aging methods.

**Strengths:**

1. Establishes a Strong Baseline: It introduces a new baseline for video re-aging, with novel contributions to architecture, dataset creation, and evaluation metrics. This provides a valuable foundation for future research in this area.
2. Demonstrates the Effectiveness of Synthetic Data: The proposed approach, while architecturally simple, effectively leverages synthetic video datasets to achieve compelling results. This highlights the potential of synthetic data for training re-aging models.
3. Provides Comprehensive Evaluation: Through extensive experiments, the authors convincingly demonstrate the realism and temporal coherence of their framework, using both qualitative and quantitative analysis.

**Weaknesses:**

1. Lack of Detail Regarding the Synthetic Dataset: The authors provide insufficient information about their synthetic dataset. To enable a comprehensive evaluation, the authors should provide detailed information about the dataset's size, diversity (including the range of ages, facial features, and other relevant attributes), and visual samples. This would allow reviewers to assess the dataset's quality and its potential impact on the reported results.
2. Missing Information on Motion Generation: Section 3.1.3 on motion generation lacks clarity regarding the stopping condition for generating intermediate frames. A more precise explanation of this process is necessary for readers to fully understand the method.
3. Unclear Availability of Resources: The authors do not explicitly state their intentions regarding the availability of the proposed dataset, pipeline code, or trained models. To enhance reproducibility and facilitate further research, it is strongly recommended that the authors publicly release these resources. Providing a link to a project page or repository, even if it's currently empty, would provide a clear indication of their commitment to open science.
4. Limited Scope of Age Progression: The generated videos primarily exhibit age-related changes in the facial area, neglecting other important regions like hair and neck skin. This inconsistency detracts from the overall realism, as subjects appear to have mismatched facial and other features.
See more detailed questions about the above weaknesses in the next section.

**Questions:**

1. Synthetic Dataset Details:
    a. Could you please provide more information about the size of your synthetic dataset, specifically the number of videos it contains?
    b. What is the average length of the generated videos in the dataset?
2. Motion Generation:
    a. In Section 3.1.3, you mention generating intermediate frames between keyframes. How many intermediate frames are typically generated?
    b. Is there a specific criterion or stopping condition that determines when to stop generating intermediate frames?
3. Spatial Masks:
    a. What is the purpose of the spatial masks M^inp and M^tar?
    b. Could you provide a visual example or description of these masks?
    c. Are they the same size as the input image I_t, and do they have the same value for all pixels?
4. Limitations in Age Progression: I noticed that the generated videos primarily show age-related changes in the facial area. Why does the proposed approach not generate changes in other regions, such as hair and neck skin? How might this limitation be addressed in future work?
5. Typos: I came across a few typos in the text, such as on line 311 and some symbols in Figure 2. Please ensure a thorough proofread to correct these errors.

---

> ### Comment · Reviewer_Lhji · 2024-12-02
> **No feedback from authors**
>
> There is no feedback or discussion from the authors. Thus, I will keep my initial rating.

---

### Official Review · Reviewer_9tWA · 2024-11-03

**Soundness:** 2
**Presentation:** 2
**Contribution:** 2
**Rating:** 3
**Confidence:** 5

**Summary:**

In this paper, the authors address the Temporal consistency issue in Video Face-Aging Approaches.
To tackle this issue, the authors introduce:
(1) A video data generation pipeline to obtain a synthetic video dataset;
(2) A video face aging framework with recurrent U-Net structure; and
(3) Temporal Regional Wrinkle Consistency (TRWC) and Temporally Age Preservation metrics to validate the temporal consistency factor as well as age transformation over time.

Experiments are employed on CelebV-HQ and VFHQ datasets to show the advantages of the proposed approach.

**Strengths:**

- The paper addresses temporal consistency factor of video face aging. This is a challenging factor in this topic.
- The paper has introduced both data generation; architecture and metrics for video face aging.

**Weaknesses:**

The novelty of the paper is limited as most sections are "inspired" or "motivated" from previous approaches.
Particularly:
- For data generation process, it relied on StyleGAN and SAM to generate aging results for single frames. Then OSFV technique is adopted to generate faces at different poses and expressions for key frames and motion generation for temporal smoothing.
- For video aging architecture, it is not novel as it is just a recurrent U-Net with commonly used losses.
The structure of the paper is more on putting multiple (previous) approaches together in an engineering manner rather than emphasizing on the novelty.

1. In line 475, "SAM fails to preserve attributes such as identity, pose and expression". However, the data generation relied on images generated by SAM (Eqn. (1)) and learn from that data. Moreover, this error will be further accumulated with OSFV method when poses and expressions presented.
- What are technical details or modifications in the proposed approach that helps to mitigate the limitation of SAM in the data generation process?
- Moreover, the authors should provide more discussions/ analysis on how the proposed approach can improve on attribute preservation. Quantitative comparison demonstrating these improvements over SAM is recommended.

2. There is no explicit constraint for consistency between frames during learning process. How can the trained network achieve the consistency when generating age-progressed frames?
Particularly:
- How can temporal consistency be enforced in the proposed approach? The authors should discuss about the details on architecture/loss functions that maintain this factor during learning/inference stages?
- Can we adopt TRWC metric as loss function for this ?

3. In Eqn. (8), why do we need to validate on the generate image rather than Delta image? In other words, can Delta images be used directly to validate the similarity rather than compute that similarity on \hat{I} and normalize with real image.
The authors should analyze on the choice of using generated images instead of delta images and its effect on the metric values. An ablation study to compare the similarity/difference between these choices is recommended.

**Questions:**

Please address the concerns in Weaknesses section

---

### Official Review · Reviewer_duBS · 2024-11-04

**Soundness:** 3
**Presentation:** 3
**Contribution:** 2
**Rating:** 3
**Confidence:** 3

**Summary:**

The paper presents a novel approach to video face re-aging, focusing on altering the apparent age of individuals in videos while maintaining temporal consistency. Key contributions include the creation of a synthetic video dataset, a baseline architecture leveraging recurrent blocks for temporal coherence, and the introduction of new metrics for evaluating age transformation quality.

**Strengths:**

1.	Innovative Data Generation Pipeline: The authors designed a comprehensive pipeline for generating a synthetic dataset specifically for model training in video face re-aging. This pipeline addresses the challenge of obtaining paired video data with consistent identities and varying ages, thereby enhancing the quality and applicability of the training data.

2.	Introduction of New Evaluation Metrics: The development of two novel metrics, Time Region Wrinkle Consistency (TRWC) and Time-Age Preservation (T-Age), provides a more effective means of assessing the quality of age transformations in videos. These metrics focus on maintaining temporal coherence, offering a more nuanced evaluation compared to traditional methods, and contributing to the advancement of the field.

**Weaknesses:**

1.	One significant shortcoming of the paper lies in its experimental section, which lacks thoroughness and depth. Specifically, the evaluation of the proposed new metrics includes only three baselines, and the quantitative comparisons in the User Study Results are limited to just two baselines. While the paper presents qualitative comparisons with various methods, these are not sufficiently persuasive without robust quantitative backing. Furthermore, the authors do not demonstrate the performance of their architecture using their own dataset to evaluate past methods, which undermines claims of superiority for their network design. This lack of comprehensive evaluation limits the credibility of the results and the overall impact of the proposed approach.
2.	Another weakness is the poor discussion of the related works. The authors merely list all current works without providing a comprehensive discussion on them. It is unclear how many methods currently exist in the video-based face re-aging area and why those methods perform poorly.
3.	The motivational section of the paper requires enhancement to better articulate the significance of the proposed method. Currently, the paper does not provide a sufficiently detailed explanation of the practical benefits and implications of the technique. The authors should elaborate on the unique challenges in video face re-aging and how their method specifically addresses these issues, thereby clarifying the motivation behind the research and its importance in the field.
4.	The paper does not explicitly address the rationale for comparing the proposed method with image-based methods. It would be beneficial for the authors to clearly state the reasons behind this comparison. The review suggests that the paper lacks an explanation of why the video approach is being contrasted with image-based techniques, and what specific advantages or insights are to be gained from this comparison. Providing this information would strengthen the paper’s argument and help the reader understand the significance of the methodological choice.

**Questions:**

1.	Did you conduct quantitative comparisons with methods such as Diffusion VAE, and did you train these methods using your own dataset to evaluate their performance? If so, could you provide the relevant experimental results and analysis?
2.	Can you add new experiments to demonstrate the effectiveness of your newly constructed dataset? For example, by using a currently common technique to conduct experiments on both the existing dataset and the dataset you provided, and using the corresponding metrics to show that your newly constructed dataset can achieve better training results.
3.	Can you clearly articulated the specific benefits of training on videos for the face age reset method, which is essential for understanding the motivation behind choosing video training over static image training? The paper seems to imply the importance of temporal consistency, but it does not explicitly state the advantages of this approach in the context of videos. Could you please elaborate on these benefits to strengthen the motivational aspect of the research and to clarify why this method is innovative and significant compared to traditional static image training methods?

---

### Meta-Review · Area_Chair_BZHX · 2024-12-21

**Metareview:**

The authors have not been able to convince three reviewers (duBS, 9tWA, azHF) towards the positive side; all these three reviewers agreed this work needs extra efforts to reach the acceptance bar of the ICLR. Thus I am inclined towards not accepting this draft at this stage. Thank you for your effort. It is an interesting work. I hope input from the reviewers will help you improve this work further.

**Additional Comments On Reviewer Discussion:**

NA

---

### Decision · Program_Chairs · 2025-01-22

Reject